# Morphing of liquid crystal surfaces by emergent collectivity

Hanne M. van der Kooij [1,2], Slav A. Semerdzhiev[1,2], Jesse Buijs [1], Dirk J. Broer [3,4], Danqing Liu[3,4] & Joris Sprakel [1]

Liquid crystal surfaces can undergo topographical morphing in response to external cues. These shape-shifting coatings promise a revolution in various applications, from haptic feedback in soft robotics or displays to self-cleaning solar panels. The changes in surface topography can be controlled by tailoring the molecular architecture and mechanics of the liquid crystal network. However, the nanoscopic mechanisms that drive morphological transitions remain unclear. Here, we introduce a frequency-resolved nanostrain imaging method to elucidate the emergent dynamics underlying field-induced shape-shifting. We show how surface morphing occurs in three distinct stages: (i) the molecular dipoles oscillate with the alternating field (10–100 ms), (ii) this leads to collective plasticization of the glassy network (~1 s), (iii) culminating in actuation of the topography (10–100 s). The first stage appears universal and governed by dielectric coupling. By contrast, yielding and deformation rely on a delicate balance between liquid crystal order, field properties and network viscoelasticity.

---

[1] Physical Chemistry and Soft Matter, Wageningen University & Research, Stippeneng 4, 6708 WE, Wageningen, The Netherlands. [2] Dutch Polymer Institute (DPI), P.O. Box 902, 5600 AX, Eindhoven, The Netherlands. [3] Stimuli-responsive Functional Materials and Devices, Department of Chemical Engineering and Chemistry, Eindhoven University of Technology, 5612 AE, Eindhoven, The Netherlands. [4] Institute for Complex Molecular Systems, Eindhoven University of Technology, 5600 MB, Eindhoven, The Netherlands. Correspondence and requests for materials should be addressed to J.S. (email: joris.sprakel@wur.nl)

In the pursuit of surfaces with programmable motility, coatings based on liquid crystal networks (LCNs) have emerged as a promising platform[1]. These coatings undergo topographical changes in response to external triggers, resulting in adaptable surface roughness, mechanics, wetting or adhesion in a pre-designed three-dimensional pattern[2–11]. Such shape-shifting LCNs have shown great potential as dynamic substrates for cell culture[10,12], on-demand self-cleaning microstructures[6,11,13], or bioinspired adhesives mimicking gecko locomotion[4,14]. To date, most attention has focused on light-actuated LC polymers, e.g. those functionalized with photoresponsive azobenzenes[2,10,11,15–21]. However, their thermal instability, low photomechanical conversion efficiency and photo-oxidative degradation have prompted the exploration of electrically active mesogens instead, which are widely employed in display panels with proven durability. The resulting polymers are capable of converting electrical energy into mechanical energy, typically in a spatially homogeneous manner or alternatively very localized[12,22–26]. To generate electrically switchable surface topographies, and thereby patterned surface properties, the electromechanical response must, however, be heterogeneous. Despite the large application potential of these coatings, only a handful have been developed in recent years[3,5,6]. Further advance of this unique class of materials would strongly benefit from a deeper understanding of the mechanisms governing topographical morphing. For example, it remains unclear how the application of an electric field sets in motion nanoscopic events that drive the ultimate microscopic shape-shifting.

Bridging the gap between the molecular level and large-scale collective motion requires a method with access to a wide range in time and length scales. While digital holography microscopy (DHM) has proven successful at visualizing surface morphing in real time with high spatial resolution[5–9], it remains superficial with limited time resolution. By contrast, conventional (polarized) optical microscopy can be very fast, yet the structural changes of deforming LC surfaces are often too subtle to resolve in detail. We therefore propose a method based on laser speckle imaging (LSI)[27–29], which offers a nanometric motion-detection resolution and allows us to spectrally decompose the complex dynamic response in a single shot; up to frequencies exceeding 10 kHz. LSI was originally devised as a medical imaging platform to visualize blood flow[30–33], and has emerged in recent years as a quantitative imaging tool to non-invasively probe the dynamics in a wide variety of complex materials[27–29,34–39]. Surprisingly, liquid-crystal-based materials are still unexplored territory.

In this paper, we establish a highly resolved view of the mechanisms underlying motility in electrically actuated liquid crystal networks. We use frequency-resolved laser speckle imaging to elucidate the complex chain of events that ultimately culminates in shape-shifting: from molecular-scale interactions between the mesogens and external field, to the emergence of patterned network expansion at the microscale. This amplification of motion in space and time is driven by collective synchronization of the mesogens, which grows towards a critical weakening of the solid network in which they are embedded. We uncover how the spatiotemporal pattern of topographical morphing can be tailored by optimizing the liquid crystal alignment and electric field properties. These insights into the inner workings of shape-shifting coatings provide clear design guidelines for the next generation of morphing surfaces.

## Results

### Surface design and shape-shifting.
We investigate the topographical morphing of well-established liquid crystal coatings which are homogeneous in chemical composition but heterogeneous in mechanical response[5]. We synthesize the LCNs by

in situ photopolymerization of a mixture of homeotropically aligned nematic liquid crystals. Half of the monomers serve as crosslinkers, while the other half are pendant on the network chains and carry a cyano group whose large dipole moment couples strongly to the field (see Fig. 1b and the Methods section). The resulting network is 2.5–3 μm thick and optically transparent, with an elastic modulus of ~2 GPa and a glass transition range between 60–120 °C (Supplementary Fig. 1)[5]. The LC coatings are applied on interdigitated indium tin oxide (ITO) electrodes, patterned onto a glass support (Fig. 1a).

While these networks are rigid and glassy at room temperature, application of an alternating current (AC) rapidly transforms the material into one that is dynamic and motile[5]. When the field is perpendicular to the LC director, the polar mesogens experience a dielectric torque and attempt to align with the field—limited only by the elastic constraints imposed by the glassy network. This partial reorientation of the mesogens gives rise to induced linear birefringence of ~0.03. As the field vector continuously changes sign, it exerts an oscillatory torque on the dangling dipoles, causing them to pivot and induce dynamic disorder that generates free volume[5,6]. In response, the network expands, resulting in substantial height modulations. Moreover, the expansion pattern can be precisely tuned by the electrode arrangement[5]. We here use interdigitated electrodes of 3 μm wide with 5 μm gaps (Fig. 1b). Surface profiling with digital holographic microscopy shows that prior to actuation, the coating is slightly corrugated with protrusions of ~50 nm that follow the contours of the electrodes (Fig. 1c, black line). 30 s after switching on the field, the surface topography is inverted (Fig. 1c, red line). The deformation persists as long as the AC field fuels it, and relaxes back to its original state within 30 s after the field is turned off (Fig. 1c, blue line). Note that dielectric heating alone cannot cause this shape-shifting, as electrothermal expansion contributes to only ~8 nm of the total deformation, thus confirming free-volume effects as the main source[5] (Supplementary Fig. 2).

### Nanoscopic imaging of surface motion.
Although DHM reveals the amplitude and direction of surface deformation, its sensitivity and resolution are not sufficient to elucidate the nanoscopic mechanisms that underlie this shape-shifting behaviour. We therefore apply a laser speckle imaging method to unravel the molecular origin of actuation in more detail[27–29]. LSI probes nanoscale motion using multiple scattering of coherent light. Since the network itself transmits nearly 100% of the visible light spectrum in both the off- and on-state[6], we induce scattering from the surface by spin-coating a thin layer of high-refractive-index $TiO_2$ pigments, and illuminate the centre of the active area with a powerful laser (Fig. 1a). We note that, due to the sample geometry, we operate in the limit of weak multiple scattering. As the photons impinge onto the sample surface, they are scattered several times, after which they travel to a camera detector. A linear polarizer filters out specular reflections and photons that have not undergone a sufficient number of scattering events. Path length differences between the photons result in an interference, or speckle, pattern on the detector. In the same experiment, we can probe both the surface expansion, derived from the absolute scattered intensity, and the surface motion, by analyzing the temporal fluctuations in speckle intensity. As the method is based on an interferometric principle, we obtain a nanometric resolution of surface deformations.

To benchmark our approach, we first measure the spatially-resolved dynamics of the LC surface described above. As the surface topography morphs, the pigment coating will follow suit; this results in an alteration of the ensemble of scattering paths and thus creates intensity fluctuations in the speckle pattern. The

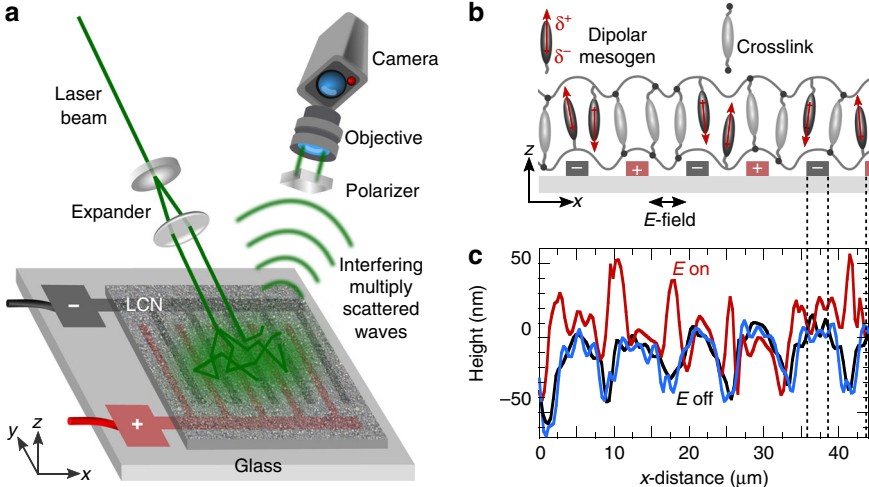

**Fig. 1** Nanoscale imaging of surface strains. **a** Schematic illustration of a laser speckle imaging (LSI) experiment. Photons from coherent plane-wave illumination, impinging on a LC device, are backscattered by $TiO_2$ pigments on the surface, and detected by a high-speed camera. Motion of the surface results in fluctuations in the detected speckle pattern, which we analyze to uncover the nanoscopic surface dynamics. **b** Schematic of the LC device prior to expansion (not to scale). Half of the mesogenic units (in dark grey) possess a permanent dipole moment which can be torqued by the field, whereas the other half (in light grey) serve as crosslinkers. **c** Surface profiles measured by digital holography microscopy (DHM): initial field-off state (black line), field-on steady state (red line), and relaxed field-off state (blue line). Switching turns the minima into maxima in a fully reversible way

amplitude and rate of these fluctuations can be used to evaluate the extent and dynamics of the surface motility. We quantify these nanoscopic motions using the contrast function $d_2$ (see the Methods section). $d_2(x, y, \Delta t, f)$ is a four-dimensional function of the spatial coordinates $x$ and $y$, time after switching on (or off) the field $\Delta t = t - t_{on}$, and a characteristic frequency $f = 1/\tau$ where $\tau$ is the lag time between correlated speckle patterns. We first fix $f$ at 1 Hz and average over 100 s after switching. Indeed, this reveals a dynamic surface texture identical to the periodicity measured by DHM (compare Fig. 2a, red line, to Fig. 1c). Between electrodes, the field lines couple strongly to the dangling dipoles and induce a large expansion, whereas on top of electrodes, the dielectric coupling is negligible and the network remains unaltered (Fig. 2a, black line). To verify whether the measured periodicity is close to the expected 8 μm of the IDE design, we take the Fourier transform (FT) of the spatial motility signal. For the $x$-direction, this indeed shows a sharp peak at ~0.123 μm$^{-1}$, denoting a periodicity of $1/0.123 = 8.1$ μm (Fig. 2b, red line). By contrast, parallel to the electrodes $d_2$ is constant, reflected by the absence of peaks in the corresponding power spectrum (Fig. 2b, black line). These data confirm that the dynamic heterogeneity originates entirely from the field pattern imprinted by the electrode arrangement.

**Elasticity-induced oscillations**. In addition to probing surface motility, LSI yields complementary insight into the surface area of the coating. To first order, the scattering intensity is proportional to the surface concentration of $TiO_2$ particles, $I \propto c_s = n/A$, which we express as the reduced change in surface area $\tilde{A} \equiv A(\Delta t)/A_0 = I_0/I(\Delta t)$. Thus, changes in intensity inversely reflect the amplitude of surface expansion. We here assume that the pigment particles adhere strongly to the network—an assumption corroborated by the fact that the original speckle pattern is largely retrieved after switching off the field, implying that surface morphing does not lead to significant rearrangement or displacement of the particles (Supplementary Fig. 3). We measure the change in surface area $\tilde{A}$ for different field frequencies $f_{field}$ and, in full agreement with digital holography[5], we find that the deformation grows with the field frequency (Fig. 3a) and is completely reversible (Fig. 3b). Moreover, the unique

sensitivity of LSI brings to light a non-monotonic change in surface area, manifest as pronounced undulations, whose frequency scales with the field frequency and decreases over time (note the logarithmic time axes and see Supplementary Fig. 3a, b). This phenomenon is reminiscent of elastic ringing. We hypothesize that upon deformation, the rigid network exerts an elastic restoring force that causes recoil and subsequent overshoot. A faster perturbation, in the form of a higher $f_{field}$, causes the surface to 'ring' at a higher frequency. This frequency decreases over time as the surface height converges to a steady state. We observe analogous oscillations when sweeping $f_{field}$, confirming the direct relationship between perturbing frequency and ringing frequency (Supplementary Fig. 4). Further evidence for the central role of surface viscoelasticity is found at elevated temperatures, where the network modulus drops[5] and accordingly the undulation period increases (Supplementary Fig. 5).

These results confirm that the overall surface area oscillates, yet to additionally prove ringing of the local surface topography, we take a closer look at the speckle pattern. Scrambling of this pattern directly reflects transformations of the surface topography. Indeed, the speckle pattern repeatedly changes and subsequently returns to almost the same texture, in an oscillatory fashion (Supplementary Fig. 6). This recoiling is also manifest as distinct echoes in the corresponding $d_2(\tau)$ traces (Fig. 3b, inset). Superimposed onto the oscillations, there is, however, a net deformation of the network, causing the echoes to eventually be lost at long $\tau$.

**Probing high-frequency molecular interactions**. Clearly, the surface morphing is determined by a complex interplay of network mechanics and dielectric interactions. To obtain a deeper understanding of the molecular mechanisms underlying the switching response, we use a novel analysis approach: Fourier-transform LSI (FT-LSI). The Fourier transform converts a signal from the time or spatial domain to the frequency domain[40,41]. Above, we used it to verify the periodicity of the motility in space; here, we apply the FT to the raw speckle intensity in time. This allows us to disentangle different temporal stages and identify dominant frequencies. We first validate our FT algorithm for a surface subjected to a linear sweep of the field frequency, which

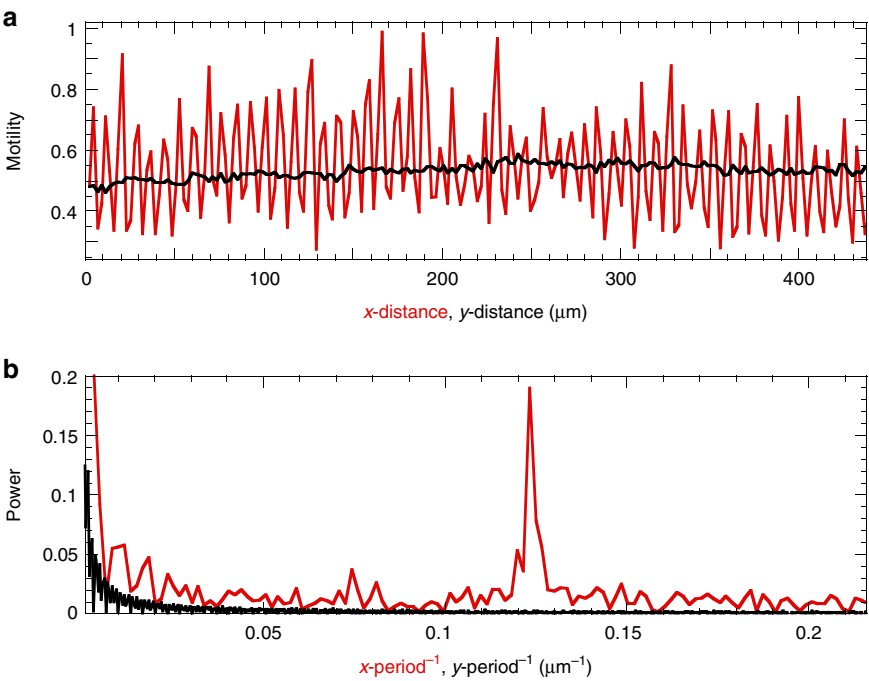

**Fig. 2** Spatial periodicity of surface motility. **a** Amplitude of the switching dynamics, probed with LSI, perpendicular (red line) and parallel (black line) to the electrodes. The motility is measured by $\langle d_2(f = 1\,\text{Hz})\rangle$, which is averaged over the orthogonal direction. The periodicity in $d_2$ matches the striated IDE pattern, as confirmed by corresponding power spectra (**b**): for the $x$-direction, a sharp peak is visible at $1/8.1\,\mu\text{m}^{-1}$, representing the distance between electrodes. By contrast, the motility along electrodes lacks clear periodicity

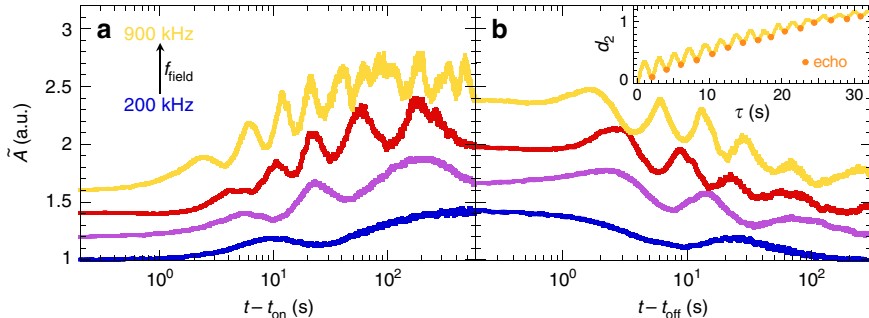

**Fig. 3** Kinetics of surface deformations. Change in surface area upon switching the field on (**a**) and off (**b**), measured by the normalized change in scattering intensity, for field frequencies of 200, 300, 500 and 900 kHz. After an initial transient, the surface gradually expands (**a**) resp. contracts (**b**), exhibiting elastic ringing. Both the undulation frequency and final deformation amplitude increase with increasing field frequency. For clarity, the curves are offset vertically by multiples of 0.2. Inset: intensity structure function $d_2$ versus lag time $\tau$. In line with elastic recoil, $d_2$ displays clear echoes (orange bullets), implying that the surface recurrently falls back to an earlier position. $t - t_{on} = 100\,\text{s}$ is chosen as reference point

should give a well-defined and sharp signal in the intensity power spectrum. Indeed, we can exactly pinpoint the driving frequency at the expected Fourier frequency (Supplementary Fig. 7). FT-LSI thus enables us to resolve even the smallest effects of dielectric coupling between an electric field and molecular dipoles.

**Temporal stages of surface morphing**. Having established the validity and sensitivity of FT-LSI, we apply our method to a surface actuated at 900 kHz. This field frequency exceeds the frame rate of the camera and thus precludes direct detection, yet we can resolve the surface dynamics ensuing from this high-frequency AC field. We again compute the intensity power spectrum as a function of time, which in this case does not exhibit a single dominant frequency i.e. straight line (Supplementary Fig. 7), but instead has a more complex dependence on frequency

and time (Fig. 4a). Notably, while DHM points to a single, monotonic change in surface height[5], LSI reveals three distinct dynamic stages (Fig. 4a). Very likely, none of these stages arises from dielectric heating, since the electrothermal expansion is slow ($<0.4\,\text{nm s}^{-1}$) and takes place at different time scales (Supplementary Fig. 2). Instead, emergent fast dynamics must be at the origin. A molecular cartoon of the three stages is shown in Fig. 4b. (I) Within tens of milliseconds after activation, the field torques the polar mesogens, and their random motion changes to pivoting motion in line with the continuously shifting field vector. The corresponding deformation is small as the mesogens oscillate individually. (II) Since the AC frequency is tuned to network resonance frequencies[5,42], the oscillations of the mesogens quickly become synchronized. This leads to cooperativity and amplification of the motion, resulting in plasticization of the network within a few seconds. The accompanying change in density causes

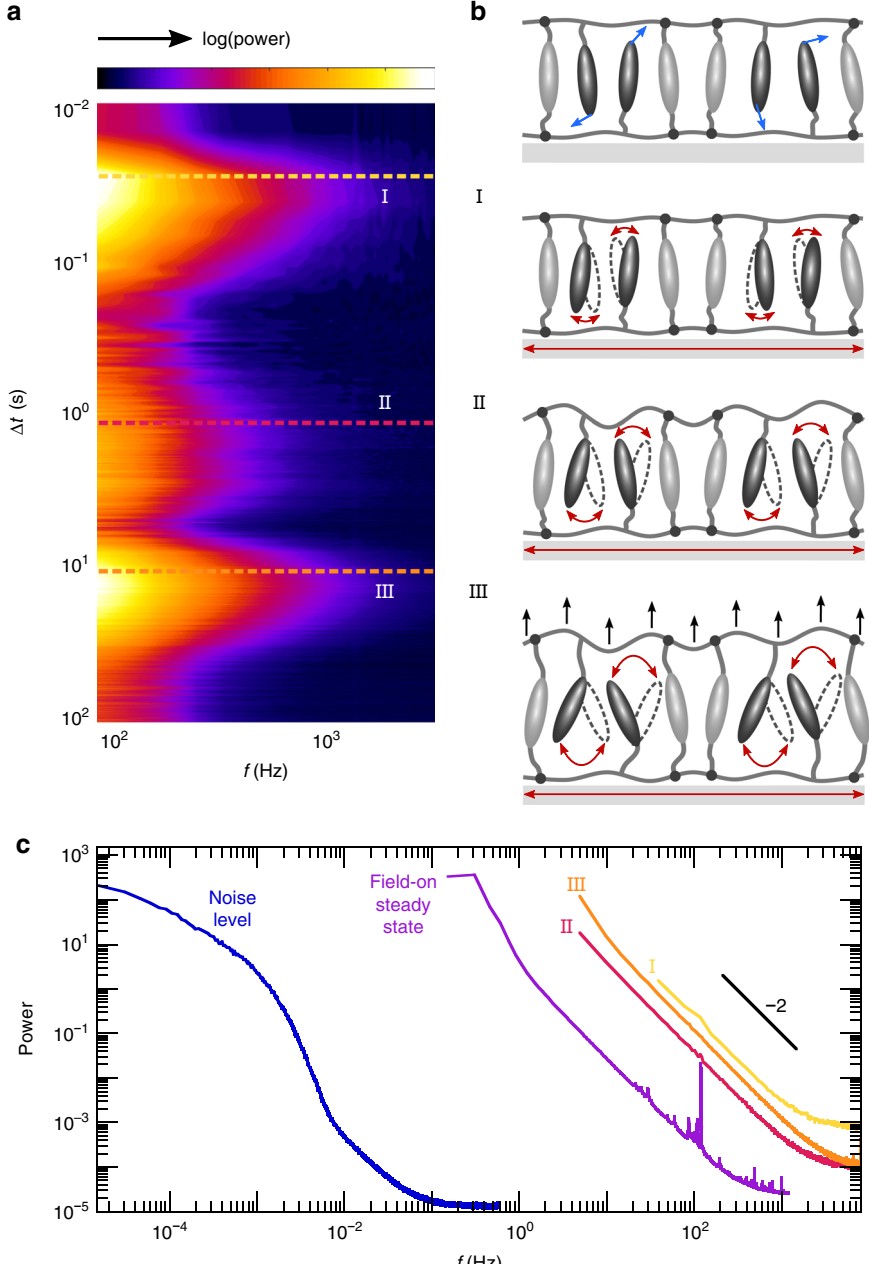

**Fig. 4** Frequency-resolved analysis of morphing kinetics. **a** Spectrogram of the dynamics after field switch-on at $\Delta t = 0$. Each horizontal strip represents one power spectrum of the temporal speckle intensity, averaged over 0.015 mm$^2$ surface. Three distinct dynamic stages can be identified. The colour scale ranges from −3 to −0.1. **b** Illustration of the field-off steady state (top) and morphing stages. Only the inter-electrode region is drawn. (I) Dielectric interactions drive the polar mesogens to oscillate along the AC field lines. (II) Under resonance conditions, the pendant and crosslink mesogens start moving cooperatively and plasticize the network. (III) A feedback loop of network weakening and increased oscillation amplitude causes amplification of the free volume, culminating in microscopic expansion. **c** Power spectra matching the indicated cross-sections in **a**. All stages exhibit high-frequency motions that are ballistic in nature i.e. characterized by power law −2. By contrast, a static reference sample (blue line) obeys a complex $f$-dependence dominated by low modes, due to external noise rather than intrinsic material properties

a transient peak in motility. (III) The plasticized network allows the mesogens to pivot with ever-increasing tilt angle, which in turn makes the network progressively weaker. This positive feedback causes an increase in free volume and resultant microscopic expansion of the surface[5,15].

Morphing of these coatings thus requires the emergence of collectivity on two levels: first, the individual motions of the dangling mesogens evolve into collective oscillations, orchestrated by the high-frequency AC field. Second, these oscillations concertedly plasticize the surrounding network, allowing also the crosslink mesogens to resonate, with steadily growing amplitude. This rising, dynamic collectivity of dangling and crosslink mesogens is amplified in both time and space, until the network elasticity brings the expansion to a halt.

To analyze the different stages in more detail, we take horizontal cross-sections of the two-dimensional power spectrum, at time points indicated by the dashed lines in Fig. 4a. Low frequencies are intrinsically most prevalent in any signal, causing a decay of power versus $f$ (Fig. 4c). The shift of this decay along the frequency axis reflects the contribution of fast dynamics to the

intensity signal. Clearly, stages I–III exhibit a large fraction of high-frequency components up to >$10^3$ Hz. By contrast, only frequencies below $10^{-1}$ Hz occur in an inert silica reference, likely due to external vibrations and slow drift. The signal we record from surfaces with emergent motility is thus more than four orders of magnitude above the noise level of our experimental set-up. Even in the steady state, $10^3$ s after switching on the field, fast nanoscale motions persist (Fig. 4c), governed by a continuous feedback loop between the field lines and dielectric properties of the network, which causes the resonance conditions to keep changing subtly. Also the shape of the power spectrum harbours information about the surface dynamics. While purely random motion would result in a power-law slope of $-3/2$, the measured slopes of $-2$ suggest ballistic motion, emphasizing the directional nature of all deformations[43,44].

**Quantification of dielectric response.** We can use the same method to evaluate in detail how the surface motility evolves over time after switching on the AC field. We do so by invoking $d_2$ as the contrast function, since we have previously shown this provides quantitative insight into mechanical deformations[29]. In this case, LSI directly measures an invariant of the surface strain tensor, i.e. the $d_2$ function represents a measure for the amplitude of the change in surface deformation as a function of time. We first compute $d_2$ for a high characteristic mechanical frequency of 100 Hz to probe the fast, transient stage I. To increase the statistical accuracy of these data, we average both in space and between the on- and off-switching response, as these are identical and symmetric (Fig. 5a–c). The dielectric origin of this stage is confirmed by the fact that its magnitude scales with the field frequency (Fig. 5a) and voltage (Fig. 5b). Moreover, its shape and rise–decay kinetics are remarkably universal for different LC topographies of similar chemical composition (Fig. 5c): a cholesteric alignment with helical structure, a planar orientation with nematic director parallel to the substrate yet perpendicular to the field lines, and an isotropic phase without order. The latter gives a weak dielectric signal because most of the mesogens do not couple efficiently to the field. All measured field frequencies, voltages and LC geometries exhibit the same kinetics in stage I: $d_2$ peaks in only ~10 ms as the field tilts the polar mesogens almost instantaneously, and subsequently falls sharply over ~200 ms as the LCN relaxes to a local energy minimum. Indeed, we are able to collapse all profiles onto a single master curve through rescaling $d_2$ with its maximum (Fig. 5d).

**Quantification of morphing dynamics.** Whereas stage I occurs under all conditions tested, the efficiency of the desired shape-shifting depends strongly on the LC orientation and field. In fact, for an isotropic LCN of the same chemical composition we detect no significant third stage (Fig. 6a, dark grey line), highlighting the crucial role of collective synchronization, which is inherently unattainable in a network without mesogen order. Below a field frequency of 40 kHz, also the homeotropic network dynamics vanish at all length scales accessible with our technique, i.e. they become sub-nanometric (Fig. 6b, blue triangles). Beyond 40 kHz, the mesogen motility and surface expansion rise rapidly as the resonance conditions are increasingly satisfied. We here take advantage of the unprecedented sensitivity of LSI, which allows detecting significantly smaller deformations than DHM. At room temperature, digital holography cannot accurately measure height changes below $f_{field} \approx 600$–700 kHz (Fig. 6b, magenta bullets)[5]. Moreover, while based on DHM the surface deformation seems complete after ~10 s (ref. [5]) LSI reveals that both the formation and relaxation of protrusions continue for hundreds of seconds (Fig. 6a, orange line). Large deformations even take hours to fully relax (Supplementary Fig. 8). Clearly, the network viscoelasticity governs the morphing kinetics and dominates over dielectric interactions, which take place on orders-of-magnitude shorter time scales[2,5].

We confirm these findings by morphing the surfaces via a different route. Instead of applying a constant 900-kHz field, we gradually increase $f_{field}$ from 0 to 900 kHz over a period of 1000 s (Fig. 6c). For the isotropic LCN, this again leads to negligible surface deformation at all field frequencies tested (dark grey line). By contrast, the homeotropic network expands rapidly with increasing $f_{field}$ (orange line), consistent with Fig. 6b, blue triangles. Interestingly, we find exactly the same threshold frequency of 40 kHz, highlighting the exceptional sensitivity of LSI, which is independent of the precise morphing kinetics and sample history.

## Discussion

In this paper, we have shown how a frequency-resolved nano-mechanical imaging technique can illuminate the hierarchical cascade of events that leads to surface morphing in liquid crystal network coatings. We uncover how an initially weak, yet measurable, dielectric response of individual mesogens to the field, amplified in time by the synchronization of pivoting motions, causes softening of the polymer matrix and finally results in network expansion and the emergence of surface topography.

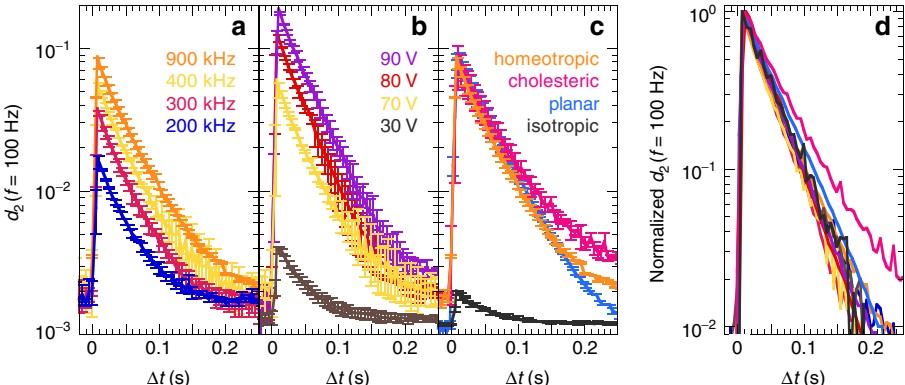

**Fig. 5** Stage I: Dielectric response. Fast transient of nanoscale motility as the field drives the polar mesogens into oscillation, whose amplitude depends on the field frequency (**a**), voltage (**b**) and LC order (**c**). The error bars are defined as s.d. of the field-on/off average. Conditions: **a** homeotropic LCN at 70 V, **b** homeotropic LCN at 400 kHz and **c** 70 V and 900 kHz. **d** Superposition of all curves through normalization highlights the dynamic signature of this stage

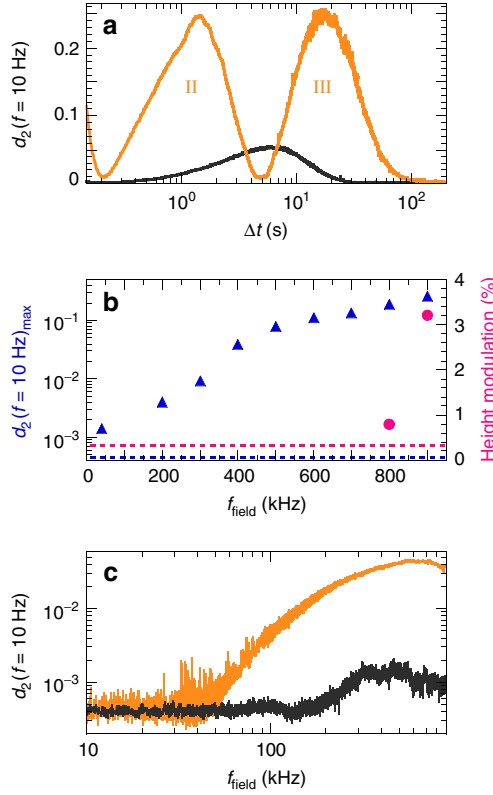

**Fig. 6 Stages II–III: Network plasticization and deformation. a** In a homeotropic LCN (orange line), the emergence of concerted mesogen dynamics (stage II) leads to efficient surface motility (stage III), whereas an isotropic network (dark grey line) lacks the final deformation stage. **b** Peak motility in stage III (blue triangles, left ordinate) and % height change measured by DHM (magenta bullets, right ordinate) versus field frequency. Both techniques show an increase in surface expansion with increasing $f_{field}$, yet the relatively lower noise level of LSI compared to DHM (dashed lines) allows probing over an order-of-magnitude lower field frequencies (down to ~40 kHz versus ~800 kHz). **c** Field frequency sweep from 0 to 900 kHz at 0.9 kHz s$^{-1}$ for a homeotropic network (orange line) and isotropic network (dark grey line), confirming the extraordinarily high displacement sensitivity of LSI

These experiments thus provide fundamental insights into the operational mechanisms by which artificial shape-shifting occurs in liquid crystal materials. Our approach paves the way to exploring how changes in molecular chemistry, mesogen alignment and device architecture influence the emergent collectivity, to ultimately optimize and amplify the shape-shifting response. Moreover, the method is readily amenable to probe these effects also in bulk materials, where similar effects can give rise to actuation and locomotion[5,12–14,16–25,45].

## Methods

**Materials.** The monomers used in this study are depicted in Fig. 7. Liquid crystal diacrylates **1** (1,4-bis-[4-{6-acryloyloxy-hexyloxy}benzoyloxy]benzene, CAS No. 125248-71-7) and **2** (1,4-bis-[4-{6-acryloyloxy-propyloxy}benzoyloxy]benzene, CAS No. 174063-87-7) form the polymer network. The cyano-capped LC monoacrylate **3** (4-[4-{6-[acryloyloxy]hexyloxy}benzoyloxy] 4-[cyano]benzoate, CAS No. 83847-14-7) couples effectively to the electric field by its large permanent dipole moment. Monomers **1** to **3** are obtained from Merck (UK). Photoinitiator **4** (phenylbis[2,4,6-trimethylbenzoyl]-phosphine oxide, CAS No. 162881-26-7) is purchased from BASF (Germany). Chiral dopant **5** (1,4:3,6-dianhydro-D-glucitol bis[4-[[4-[[[4-[{1-oxo-2-propenyl}oxy]butoxy]carbonyl]oxy]benzoyl]oxy]benzo-ate], CAS No. 223572-88-1) is obtained from Ciba (Switzerland) and added in small amount to induce a cholesteric (chiral nematic) phase with helical pitch of ~450 nm, only used for Fig. 5c, d. All samples are prepared from a mixture of 24.75 wt% monomer **1**, 24.75 wt% monomer **2**, 49.5 wt% monomer **3** and 1.0 wt%

photoinitiator **4**, dissolved in dichloromethane (CAS No. 75-09-2, purchased from Sigma–Aldrich, USA). Polyimide 7511L SUNEVER is obtained from Nissan Chemical Corporation (Japan), and polyimide AL-1051 from JSR Corporation (Japan). TiO$_2$ nanoparticles (~40 nm diameter, coefficient of variation ~40%, purity 99.5%) are purchased from US Research Nanomaterials (USA). All components of the LSI set-up are purchased from Thorlabs (Germany), unless otherwise specified.

**Sample preparation.** To establish the desired LC alignment, the glass substrates with patterned ITO are first cleaned and spin-coated with polyimide, followed by baking to yield a polyimide film thickness of ~30 nm. Polyimide 7511L is applied to obtain homeotropic alignment of the LC monomer mixture, and polyimide AL-1051 to achieve planar (parallel) anchoring to the substrate. The latter is additionally rubbed with a polyester cloth after baking to orient the nematic director along the y-direction i.e. perpendicular to the field lines. Subsequently, a thin film of the monomer mixture is formed by spin-coating from solution, and photo-polymerized by UV light that excites photoinitiator **4**. The UV exposure is continued for 5 min under N$_2$ using a mercury lamp (OmniCure S2000, Lumen Dynamics Group Inc., Canada) at 26 °C to lock a (chiral) nematic orientation or at 70 °C to obtain an isotropic LCN. The samples are post-baked at 120 °C under N$_2$ to ensure complete cure of the acrylate monomers. The resulting LCNs are transparent and have order parameters between 0.6 and 0.7, with the exception of the disordered isotropic network. The nematic and isotropic networks show no domain formation between crossed polarizers. To render the coatings multiply scattering for LSI experiments, a thin layer of high-refractive-index TiO$_2$ (titania) nanoparticles is deposited onto the surface by spin-coating a 1 wt% aqueous TiO$_2$ suspension at 2000 rpm for 30 s. The suspension is sonicated for 10 min prior to spin-coating to ensure that any aggregated particles are well-dispersed. The spin-coating conditions are fine-tuned to render the coating intermediately scattering, i.e. a well-developed speckle pattern is obtained through crossed polarizers implying that the scattered photons are sufficiently randomized, yet TiO$_2$ coating is thin enough to not interfere with the surface morphing.

**Characterization.** An in-plane, sinusoidal electric field is generated by a function generator (TGA1241, TTi Inc., USA) connected to an amplifier (F20A, FLC Electronics, Sweden). Unless otherwise specified, the LCN coating is actuated at 25 °C at a peak-to-peak voltage of 70 V (i.e. a field strength of 14 V/μm) and a field frequency of 900 kHz. The output AC signal is monitored with an oscilloscope (TBS1022, Tektronix, USA). The surface profiles in Fig. 1c are measured using digital holography microscopy (Lyncée Tec SA, Switzerland), as described extensively in ref. [5]. The current-induced temperature change of the sample surface is measured using an infrared sensor (PCE-IR 51, PCE Instruments, Germany).

**Laser speckle imaging.** See ref. [28] for a detailed description of the LSI method and our custom-built set-up, which we here operate in the intermediate scattering regime rather than in the strong multiple scattering limit. In brief: the active area of the LC surface is illuminated with an expanded coherent laser beam (Cobolt Samba, 1 W, λ = 532 nm, Cobolt, Sweden), whose photons are weakly multiply scattered by the TiO$_2$ nanoparticles on the surface. As a result of these scattering events, the photon paths approximate a diffusive rather than a ballistic trajectory. Each photon traverses a different, unique path, leading to path length differences among the photons that cause spatially random constructive and destructive interference in the scattered light pattern. This so-called speckle pattern is recorded on a camera, which is here in the backscatter geometry. Only the multiply scattered light is detected, with specular and low-order scattering paths filtered by a linear polarizer perpendicular to the polarization of the incident laser beam. Two cameras are alternately used: a Dalsa Genie CCD camera (CR-GM00-H6400, Stemmer Imaging, Netherlands) for continuous streaming at acquisition rates up to 200 fps, and a HiSpec 1 CMOS camera (Fastec Imaging, USA) for imaging at acquisition rates up to 40,000 fps.

As the surface-attached nanoparticles move, in response to internal sample dynamics, all photon path lengths change, causing the speckle pattern to change accordingly. The rate of these intensity fluctuations is quantified on a pixel-by-pixel basis using the intensity structure function $d_2$:

$$d_2(x,y,\Delta t, 1/f) = \frac{\langle [I(x,y,\Delta t) - I(x,y,\Delta t + 1/f)]^2 \rangle}{\langle I(x,y,\Delta t) \rangle \cdot \langle I(x,y,\Delta t + 1/f) \rangle} \qquad (1)$$

where $I$ is the speckle intensity at position x,y and time after switching on or off the field $\Delta t$, and $f = 1/\tau$ with $\tau$ the time separating the two compared speckle patterns. The angular brackets denote averaging in time and/or space. If the surface is static over the lag time $\tau$, the speckle pattern will be unaltered and hence $d_2$ is zero. The more dynamics have occurred during $\tau$, the faster the speckle intensity fluctuates and the larger the numerator will be. The analysis frequency $f$ thus serves as a means to distinguish processes occurring on different time scales. By using symmetric normalization in the denominator, the magnitude of $d_2$ becomes independent of the absolute speckle intensity and hence is universal regardless of the optical properties and geometry. Note that the intermediate scattering regime limits quantitative analysis beyond $d_2$, since rigorous theory is absent for this regime. Nevertheless, $d_2$ itself can be interpreted in a quantitative and reproducible manner. The intermediate scattering geometry moreover combines the best of both

**Fig. 7** Chemical structures of the monomers used in the LCN syntheses

multiple and single scattering, unifying high sensitivity with applicability to thin films.

Fourier-transform LSI is a new, complementary approach that allows resolving the entire frequency spectrum exceptionally fast. It is based on the short-time fast Fourier transform (ST-FFT) of the temporal intensity signal[40]. Using a custom-written Matlab routine, the power spectrum is computed for every pixel as a function of time and subsequently averaged over space. Although the absolute frequencies in the power spectrum and $d_2$ function cannot be compared one-to-one, the dependencies of power and $d_2$ on $f$ are related, according to the Wiener–Khinchin theorem[41]. We therefore use the two interchangeably, with $d_2$ a well-established LSI parameter yet the power spectrum orders of magnitude faster in computing time. More detailed information about the Fourier analysis is provided in Supplementary Note 1.

## Data availability
The data that support the findings of this study are available from the corresponding author upon reasonable request.

## Code availability
The Matlab code and mathematical algorithms used to process and analyze the raw data are available from the corresponding author upon reasonable request.

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

## Acknowledgements

The research of H.M.v.d.K. and S.A.S. forms part of the research programme of the Dutch Polymer Institute (DPI), projects #781 and #913ft16. The work of J.B. was carried out as part of a project of the Institute for Sustainable Process Technology: Controlling Multi Phase Flow (WP-30-01). The contribution of D.J.B. was supported by the European Research Council with the ERC Advanced Grant 66999 (VIBRATE). The work of D.L. is part of the NWO VENI research programme with project number 15135, and that of J.S. of the NWO VIDI research programme with project number 723.016.001. Merck is acknowledged for providing ITO IDE.

## Author contributions

H.M.v.d.K. and J.S. developed the methodology, D.J.B. and D.L. designed and prepared the devices, H.M.v.d.K., S.A.S. and D.L. performed the experiments, J.B. developed FT-LSI, H.M.v.d.K. analyzed the data, all authors designed the study, discussed the data and prepared the manuscript.

## Additional information

**Competing interests:** The authors declare no competing interests.

