## [peer review file · Nature Communications]

Reviewer #1 (Remarks to the Author):

This article describes investigation of nanoscopic mechanism on topological morphing of liquid crystal surfaces induced by an electrical stimulus. The authors use laser speckle imaging (LSI) to detect manometric motion and spectrally decompose the complex dynamic response. The novelty of this article is that they introduce this method to observe the dynamic surface of liquid crystal elastomers.

The elucidation of surface motion that start with nanoscale molecular motion and becomes concertedly amplified in microscale motion is indeed important from the viewpoint of high performance materials design. Spatiotemporally comprehensive understanding of the phenomena is required, which naturally remains challenge.

The results obtained by LSI revealed that an amplifying process of nanoscale molecular motion to surface deformation with three stages from 10 ms to 100s. This was never achieved by the conventional digital holographic microscopy (DHM).

This work seems to give an impact to the community if LSI is applicable to topological morphing liquid crystal elastomers, by convincing with further evidences after the following questions need to be considered.

1. LSI detects speckle changes by light scattering from TiO₂ particles. How do the authors prove that speckle changes reflect surface morphing of liquid crystal elastomers? Liquid crystals with high order parameters could scatter incident light by surface morphing, order-disorder, and molecular fluctuation.

2. Detailed information of TiO₂ particles needs to be shown. What are size, coefficient of variation, surface density? Is assumption that particles are adhered at the same point of the elastomer surface acceptable?

3. Three stages simply lead to in-plane molecular orientation between electrodes. Polarized optical microscopy must enable us to observe optical anisotropy by electrical field application. Also, induced birefringence could be estimated, which gives the degree of orientational changes. There must be some relationship between a free-volume increase and molecular motion.

4. Crosslinked liquid crystal elastomers generally undergo contraction when homeotropic alignment becomes disordered. Does LSI distinguish surface elevation and depression? Did the authors first assume surface elevation by an electric field and estimate how much the surface was moved?

5. Figure 7 clearly shows that LSI can detect surface morphing at smaller frequency. At 800 and 900 kHz, the values obtained by DHM and LSI are different. How is the reliability of LSI (or DHM) confirmed with morphing of liquid crystal elastomer surfaces?

Reviewer #2 (Remarks to the Author):

van der Kooij et al. describe the use of a nanoscale strain imaging process to describe the dynamics of AC electric field deformations in glassy liquid crystal networks. This work provides the most detailed description of this mechanism of deformation, and this detailed understanding will be critical to advancing understanding in this relatively new area of smart polymers. The conclusions of this work are largely supported by the data. The references and description of the methods used are largely appropriate. However, there are a few minor concerns that should be addressed prior to publication.

1. The discussion on elastic ringing is relatively weakly supported by the presented data. Is any evidence available that the timescale of the ringing is reasonable?

2. The elastic ringing argument is partially supported by measurements at several temperatures. At higher temperatures (and presumably higher damping) conditions the ringing vanishes. Notably, as this ringing is at large timescales it would be helpful to have dynamic mechanical measurements of elastic modulus and tan delta at correspondingly low frequencies. At a minimum, the discussion should be modified to describe this effect more thoroughly.

3. Are the networks crosslinked in the isotropic phase actually isotropic or do they have a polydomain texture? The pitch of the chiral nematic should be mentioned.

4. Only a single chiral composition is described in the methods, but a film with planar orientation is described in figure 6.

Reviewer #3 (Remarks to the Author):

The authors introduce a new mechanism of topographical morphing in stimuli-sensitive liquid-crystalline networks (LCNs). In brief, these glassy networks undergo plasticization, as dangling mesogens serve as molecular dipoles that oscillate in an alternating magnetic field. These oscillations serve to increase the free volume of the network and enable increase polymer chain mobility, resulting in topological changes that correspond to the geometry of patterned electrodes that rest underneath the sample. To measure this phenomenon, laser-speckle imaging (LSI) and Fourier-Transform LSI were developed and employed. The study is well designed and systematic in investigating this phenomenon. I only have minor comments for the authors to address before recommending for publication.

A fundamental premise of this study is that the oscillating molecular dipoles cause plasticization of the network. The authors report that the network has a glass transition that ranges from 60 to 120 Celsius; however, they do not show or explain how the glass transition was characterized. While I do not doubt that the authors know how to characterize the glass transition, it would be helpful for the reader to see and verify these indeed are glassy networks at room temperature and the small amount of electro-thermal heating would not cause an increase in chain mobility and free volume.

The authors should comment further on the role of liquid crystal order and shape switching. The study conclusively shows that liquid-crystal order is needed to effectively couple the electric field with the mesogens to induce oscillations (i.e., the isotropic networks demonstrated a negligible effect). LCNs and liquid-crystal elastomers are well known for their two-way actuation behavior due to isotropic transition that occurs when mesogen order is disrupted. The authors should comment further regarding if the mesogens, while oscillating and resonating, are believed to maintain or lose their order. The authors attribute much of the topological morphing to free volume effects, but I am inclined to believe it is a combination of free volume and a change in order parameter that help drive the shape switching. The authors state that the networks will return to their original shape after 30 seconds of turning off the electric field. When studying amorphous (non-liquid-crystalline) networks, reductions in free volume typically take on the order of days to relax in the glass state (as seen when a networked is quenched below its glass transition and held indefinitely). The authors should comment on if the homeotropic order of the mesogens helps accelerate the collapse in free volume and helps return the polymer to its original state.

My last comment is that the term "emergent collectivity" is only used once in the manuscript in the Discussion section. Since this term is used in the title and used to describe this phenomenon, it should get a proper definition and discussion within the text.

Minor formatting and manuscript suggestions:

Figure 4 and the section on Probing High-Frequency Molecular Interactions seems more like a validation of the technique, rather than a result. It may be better suited in the supplemental information.

-The CAS numbers for each material should be provided.

-The names for the mesogens should be provided.

-In the Methods section, suppliers are inconsistently mentioned. Sometimes the manufacturer and country of the manufacturer are omitted.

Reviewer #1 (Remarks to the Author):

This article describes investigation of nanoscopic mechanism on topological morphing of liquid crystal surfaces induced by an electrical stimulus. The authors use laser speckle imaging (LSI) to detect manometric motion and spectrally decompose the complex dynamic response. The novelty of this article is that they introduce this method to observe the dynamic surface of liquid crystal elastomers.

The elucidation of surface motion that start with nanoscale molecular motion and becomes concertedly amplified in microscale motion is indeed important from the viewpoint of high performance materials design. Spatiotemporally comprehensive understanding of the phenomena is required, which naturally remains challenge.

The results obtained by LSI revealed that an amplifying process of nanoscale molecular motion to surface deformation with three stages from 10 ms to 100s. This was never achieved by the conventional digital holographic microscopy (DHM).

This work seems to give an impact to the community if LSI is applicable to topological morphing liquid crystal elastomers, by convincing with further evidences after the following questions need to be considered.

We thank the reviewer for his/her positive words about the relevance and novelty of the results described in our manuscript. We highly value his/her constructive and detailed comments that have helped us considerably to improve our manuscript. We hope that the discussion below will further clarify the reviewer's comments.

1) LSI detects speckle changes by light scattering from TiO₂ particles. How do the authors prove that speckle changes reflect surface morphing of liquid crystal elastomers? Liquid crystals with high order parameters could scatter incident light by surface morphing, order-disorder, and molecular fluctuation.

During all stages of actuation, our liquid crystal networks remain optically transparent, and consequently do not scatter the incident laser light to a significant extent. Only when coated with TiO₂ nanoparticles, our samples scatter sufficient light to generate a speckle pattern on the camera. Changes in this speckle pattern must ensue from surface morphing, as the other two options are ruled out: although there is indeed a minor decrease in order parameter after switching on the field, it does not change the network turbidity i.e. it takes place on very small length scales, and the same is applicable to molecular fluctuations. Using our LSI approach, we do however observe the *result* of order-disorder transitions and molecular fluctuations in the form of displacements of the TiO₂ nanoparticles, which directly reflect deformations of the coating surface. In our revised manuscript, we have emphasized the optical transparency of the LCN (i.e. the absence of scattering from anything but the TiO₂ particles) at the top of p. 6, supported by reference 6.

2) Detailed information of TiO₂ particles needs to be shown. What are size, coefficient of variation, surface density? Is assumption that particles are adhered at the same point of the elastomer surface acceptable?

The reviewer is fully correct. In the Materials subsection of the Methods (p. 17), we have added more information about the TiO₂ particles. These particles indeed adhere strongly to the coating surface. For all on-off cycles of the field tested, we have found that after switching off the field, the speckle pattern returns largely to its initial state, implying that surface morphing does not lead to significant rearrangement or displacement of the TiO₂ particles. To highlight this, we have included three typical

speckle patterns in Supplementary Fig. S3, and addressed it in the corresponding caption and main text at the bottom of p. 8.

It is very difficult to assess the surface density of the TiO₂ particles. Such information would indeed be required if we pursued microrheology-type analysis. However, our aim was not to quantify the exact surface deformation, but to obtain a measure for the *change* in surface deformation as a function of time and different parameters. The demands for the optical properties are less stringent in this case. Hence, rather than attempting to obtain a perfect monolayer of monodisperse pigments at the LCN surface, we have optimized the spin-coating conditions such that our samples fulfilled the intermediate-scattering requirements. This means that we acquired a well-developed speckle pattern through crossed polarizers (implying that the scattered photons were sufficiently randomized) yet the TiO₂ coating was thin enough to not interfere with the surface morphing (i.e. the detected scattering intensity was polarization dependent, implying that the surfaces were not highly multiple scattering). We have explained this approach better in the revised manuscript, at the end of the subsection 'Sample preparation' in the Methods, p. 18–19.

3) Three stages simply lead to in-plane molecular orientation between electrodes. Polarized optical microscopy must enable us to observe optical anisotropy by electrical field application. Also, induced birefringence could be estimated, which gives the degree of orientational changes. There must be some relationship between a free-volume increase and molecular motion.

The reviewer has an excellent suggestion. The surface expansion is indeed based on free volume generation as a result of the dynamics of the molecular rods deflecting from their initial alignment. In the case of the homeotropic sample, some in-plane birefringence is built up, due to the biased directional change of the order. This induced birefringence is of the order of 0.03. We have included a sentence on this in the second paragraph on p. 5.

4) Crosslinked liquid crystal elastomers generally undergo contraction when homeotropic alignment becomes disordered. Does LSI distinguish surface elevation and depression? Did the authors first assume surface elevation by an electric field and estimate how much the surface was moved?

We agree that contraction along the director is indeed observed for unconstrained liquid crystal networks, in which case expansion occurs only perpendicular to the director. In the case of LCNs that are firmly connected to their substrate, this would lead to shear, which might become visible at the edges of director change of director-patterned coatings, or at a topological defect. However, in our current work we study a continuous, mono-domain, homeotropic LCN, whose expansion in lateral direction is highly limited because of substrate attachment and inter-electrode confinement. The increase in volume thus inevitably 'escapes' into the z-dimension perpendicular to the substrate.

The reviewer is correct that LSI cannot differentiate between surface elevation and depression. Nevertheless, the abovementioned argument strongly points to field-induced elevation, which is confirmed by DHM. Digital holography and laser speckle imaging are thus complementary techniques: DHM provides absolute numbers and directionality of height changes, while LSI detects much smaller deformations and thereby the governing mechanisms. The combination of the two techniques provides the most complete picture. We have emphasized the power of DHM at the top of the last paragraph on p. 5.

5) Figure 7 clearly shows that LSI can detect surface morphing at smaller frequency. At 800 and 900 kHz, the values obtained by DHM and LSI are different. How is the reliability of LSI (or DHM) confirmed with morphing of liquid crystal elastomer surfaces?

Although the values acquired using LSI and DHM indeed seem different, the two techniques are not in contradiction, as they measure different quantities. The apparent discrepancy in Fig. 6b (former Fig. 7b) is rooted in the much higher displacement resolution of LSI compared to DHM. The two y-axes thus cannot be compared quantitatively.

The validity of our LSI technique in general has been proven for various materials, as demonstrated in references 27–29. We believe that its reliability concerning the morphing of LCNs is verified by the highly similar trends we find using DHM, e.g. in the spatial periodicity (compare Fig. 1c with 2a), field frequency dependence (compare Fig. 5a/6b with reference 5), and field strength dependence (compare Fig. 5b with reference 5). DHM itself is a well-established technique which is widely used for studying surface deformation. It has also been combined with AFM, ellipsometry and friction measurements, all confirming its reliability.

Reviewer #2 (Remarks to the Author):

van der Kooij et al. describe the use of a nanoscale strain imaging process to describe the dynamics of AC electric field deformations in glassy liquid crystal networks. This work provides the most detailed description of this mechanism of deformation, and this detailed understanding will be critical to advancing understanding in this relatively new area of smart polymers. The conclusions of this work are largely supported by the data. The references and description of the methods used are largely appropriate. However, there are a few minor concerns that should be addressed prior to publication.

We are thankful for the reviewer's positive assessment of our work. We have carefully taken into account his/her comments, as outlined below.

1) The discussion on elastic ringing is relatively weakly supported by the presented data. Is any evidence available that the timescale of the ringing is reasonable?

The elastic ringing of our coatings is a complex phenomenon, which cannot be captured by a single timescale, but instead depends on the applied field frequency and the (continuously evolving) network properties. However, our observed trends of the ringing frequency versus f_{field} and Δt are in line with expectations. First of all, the ringing frequency increases with increasing field frequency, both for a constant f_{field} (Fig. 3) and for an f_{field} sweep (Supplementary Fig. S4). Second, the ringing frequency strongly decreases over time after actuation, which indicates a gradual convergence of the surface height towards a steady state (see Fig. 3 and Supplementary Fig. S3a,b). Immediately after switching on the field, the network modulus, density and temperature are fluctuating strongly, thus causing fast recoil and overshoot of the surface as it fails to meet the rapidly changing resonance conditions. Over time, the field frequency and network dynamics become increasingly synchronized, allowing the ringing to subside.

2) The elastic ringing argument is partially supported by measurements at several temperatures. At higher temperatures (and presumably higher damping) conditions the ringing vanishes. Notably, as this ringing is at large timescales it would be helpful to have dynamic mechanical measurements of elastic modulus and tan delta at correspondingly low frequencies. At a minimum, the discussion should be modified to describe this effect more thoroughly.

The reviewer is absolutely correct that we should elaborate on the elastic ringing. We have accordingly discussed this phenomenon in more detail in the revised manuscript, last two paragraphs on p. 9 and top of p. 10. Additional evidence for our hypothesis is included as Supplementary Fig. S6. Furthermore, we have added DMTA results for the homeotropic topography in Supplementary Fig. S1.

3) Are the networks crosslinked in the isotropic phase actually isotropic or do they have a polydomain texture? The pitch of the chiral nematic should be mentioned.

The networks crosslinked in their isotropic phase appear fully black between crossed polarizers, and no domain formation is seen. Indeed, we performed the crosslinking far away from conditions causing domain formation of random molecular order, i.e. we ensured that i) the polymerization proceeded fast, ii) at a temperature far above the nematic-to-isotropic transition, and iii) with high degree of crosslinking. The sample with cholesteric order has a pitch of approximately 450 nm. We have thankfully included these important aspects in our revised manuscript, first paragraph on p. 17 and end of p. 18.

4) Only a single chiral composition is described in the methods, but a film with planar orientation is described in figure 6.

The composition of the planar sample is identical to that of the homeotropic sample, but the alignment was forced to be parallel to the substrate by a thin polyimide film rubbed prior to polymerization. This information was somewhat hidden, and is consequently clarified in the revised manuscript, subsection 'Sample preparation' of the Methods, p. 17–18.

Reviewer #3 (Remarks to the Author):

The authors introduce a new mechanism of topographical morphing in stimuli-sensitive liquid-crystalline networks (LCNs). In brief, these glassy networks undergo plasticization, as dangling mesogens serve as molecular dipoles that oscillate in an alternating magnetic field. These oscillations serve to increase the free volume of the network and enable increase polymer chain mobility, resulting in topological changes that correspond to the geometry of patterned electrodes that rest underneath the sample. To measure this phenomenon, laser-speckle imaging (LSI) and Fourier-Transform LSI were developed and employed. The study is well designed and systematic in investigating this phenomenon. I only have minor comments for the authors to address before recommending for publication.

We thank the reviewer for his/her positive evaluation of our manuscript, for the critical reading and helpful comments.

1) A fundamental premise of this study is that the oscillating molecular dipoles cause plasticization of the network. The authors report that the network has a glass transition that ranges from 60 to 120 Celsius; however, they do not show or explain how the glass transition was characterized. While I do not doubt that the authors know how to characterize the glass transition, it would be helpful for the reader to see and verify these indeed are glassy networks at room temperature and the small amount of electro-thermal heating would not cause an increase in chain mobility and free volume.

We agree with the reviewer that such information is relevant, and we have therefore added dynamic mechanical thermal analysis data for the homeotropic topography in Supplementary Fig. S1.

2) The authors should comment further on the role of liquid crystal order and shape switching. The study conclusively shows that liquid-crystal order is needed to effectively couple the electric field with the mesogens to induce oscillations (i.e., the isotropic networks demonstrated a negligible effect). LCNs and liquid-crystal elastomers are well known for their two-way actuation behavior due to isotropic transition that occurs when mesogen order is disrupted. The authors should comment

further regarding if the mesogens, while oscillating and resonating, are believed to maintain or lose their order. The authors attribute much of the topological morphing to free volume effects, but I am inclined to believe it is a combination of free volume and a change in order parameter that help drive the shape switching. The authors state that the networks will return to their original shape after 30 seconds of turning off the electric field. When studying amorphous (non-liquid-crystalline) networks, reductions in free volume typically take on the order of days to relax in the glass state (as seen when a networked is quenched below its glass transition and held indefinitely). The authors should comment on if the homeotropic order of the mesogens helps accelerate the collapse in free volume and helps return the polymer to its original state.

When the liquid crystal network is actuated, indeed some loss of order takes place, resulting in induced birefringence of the order of 0.03 (see our reply to comment 3 of reviewer 1). However, it is in fact this loss of order – when occurring in a high-frequency oscillation – which leads to volume generation, as we have proven in references 2 and 5. Both light and an electrical field can do this. The free volume is maintained only under the condition that it is continuously addressed at the right field frequency. At too low frequencies, the created volume is directly taken by other parts of the network; this process takes place on a time scale of seconds (also here, a similarity exists between light- and electric-field-actuated systems). This free volume induced by network oscillation deviates from the classical free volume observed in polymer networks which is obtained by quenching from the rubbery to the glassy state. Only at large surface deformations, we observe additional relaxation on a time scale of hours, more related to the classical relaxation.

We thank the reviewer for this insight, and have accordingly included Supplementary Fig. S8 to reveal this long-term relaxation. In the main text, we have added a sentence about this at the top of p. 16.

3) My last comment is that the term "emergent collectivity" is only used once in the manuscript in the Discussion section. Since this term is used in the title and used to describe this phenomenon, it should get a proper definition and discussion within the text.

We fully agree with the reviewer, and have dedicated a paragraph to this phenomenon (2nd paragraph on p. 12).

4) Minor formatting and manuscript suggestions:

Figure 4 and the section on Probing High-Frequency Molecular Interactions seems more like a validation of the technique, rather than a result. It may be better suited in the supplemental information.

We have moved the figure and corresponding text to the Supplementary Information, Fig. S7.

- The CAS numbers for each material should be provided.

This is an excellent suggestion. We have provided the CAS numbers in the Materials section of the Methods, p. 17.

- The names for the mesogens should be provided.

We have provided the names of the five monomers in the Materials section of the Methods, p. 17.

- In the Methods section, suppliers are inconsistently mentioned. Sometimes the manufacturer and country of the manufacturer are omitted.

The reviewer is correct. We have completed all the missing information on p. 17–19.

Reviewer #1 (Remarks to the Author):

Comments and revision of the manuscript based on the previous comments seem to be convincing and fair, and I am satisfied by the revised manuscript. I recommend publication of the manuscript in Nat. Common. as is.

Reviewer #2 (Remarks to the Author):

The authors have addressed the concerns of each of the reviewers. In my opinion, this work will be impactful in the area of responsive materials. I recommend publication.

Reviewer #3 (Remarks to the Author):

The authors have addressed my concerns. I would recommend for publication.

Reviewer #1 (Remarks to the Author):

Comments and revision of the manuscript based on the previous comments seem to be convincing and fair, and I am satisfied by the revised manuscript. I recommend publication of the manuscript in Nat. Common. as is.

Reviewer #2 (Remarks to the Author):

The authors have addressed the concerns of each of the reviewers. In my opinion, this work will be impactful in the area of responsive materials. I recommend publication.

Reviewer #3 (Remarks to the Author):

The authors have addressed my concerns. I would recommend for publication.

Response to reviewers

We thank the reviewers for assessing our revised manuscript. Considering the positive evaluation by all three reviewers after the first round of revisions, with no further remarks for improvement or new questions raised, we keep the manuscript in its current form with respect to the scientific contents.